


# Comparison of mesospheric sodium profile retrievals from OSIRIS and SCIAMACHY nightglow measurements

Julia Koch[1], Adam Bourassa[2], Nick Lloyd[2], Chris Roth[2], and Christian von Savigny[1]

[1]Institute of Physics, University of Greifswald, Greifswald, Germany
[2]University of Saskatchewan, Saskatoon, SK, Canada

**Correspondence:** Julia Koch (j.koch.physik@uni-greifswald.de)

**Abstract.** Sodium airglow is generated when excited sodium atoms emit electromagnetic radiation while they are relaxing from an excited state into a lower energetic state. This electromagnetic radiation, the two sodium D-lines at 589.0 nm and 589.6 nm, can usually be detected from space or from ground. Sodium nightglow occurs at times when the sun is not present and excitation of sodium atoms is a result of chemical reaction with ozone. The detection of sodium nightglow can be a means

to determine the amount of sodium in the earths' mesosphere and lower thermosphere (MLT). In this study, we present time series of monthly mean sodium concentration profiles, by utilizing the large spatial and temporal coverage of satellite sodium D-line nightglow measurements. We use the OSIRIS/Odin mesospheric limb measurements to derive sodium concentration profiles and vertical column densities and compare those to measurements from SCIAMACHY/Envisat and GOMOS/Envisat. Here we show that the Na D-line Limb Emission Rate (LER) and Volume emission rate (VER) profiles calculated from the

OSIRIS and SCIAMACHY measurements, although the OSIRIS LER and VER profiles are around 25 % lower, agree very well in shape and overall seasonal variation. The sodium concentration profiles also agree in shape and magnitude, although those do not show the clear semi-annual cycle which is present in the LER and VER profiles. The comparison to the GOMOS sodium vertical column densities (VCD) shows that the OSIRIS VCDs are in the same order of magnitude although again the semi-annual cycle is not as clear. We attribute the differences in the LER, VER and sodium profiles to the differences in spatial

coverage between the two satellite measurements, the lower signal-to-noise ratio (SNR) of the SCIAMACHY measurements and differences in local time between the measurements of the two satellites.

## 1 Introduction

It has been known for a long time that the night sky, even in moonless nights, is sometimes not completely dark. In 1868, Anders Ångstrom, who is the namesake of the Ångstrom unit, noticed that the color of the night time illumination was the same as

the color of aurora, although the highly structured features were clearly missing (Ångström, 1869). About 50 years later, in the 1920s, John McLennan and G. M. Shrum found the green emission line of oxygen which could explain the two apparently related phenomena in the night sky (McLennan and Shrum, 1925). Less than a decade later, V.M. Slipher found that in addition to the oxygen green line, the night sky emissions contained features at wavelengths corresponding to the sodium doublet. Therefore, he concluded that there has to be a layer of sodium atoms in the upper atmosphere (Slipher, 1929). Today, we know,





that the sodium layer is located in the Mesosphere and Lower-Thermosphere (MLT region), which is found at altitudes between 70 and 110 km. In that layer sodium concentrations range between ca. 500 and ca. 5000 $atoms/cm^3$ depending on altitude, location, and both the time of the day and the time of the year. A very prominent feature of the sodium layer is its semiannual cycle in the tropics with peaks around the spring and fall equinoxes (Takahashi et al., 1995). This has also been observed in other atmospheric parameters, e.g. OH rotational temperature, OH emission rate, oxygen green line emission rate and atomic

oxygen concentration (e.g., von Savigny and Lednyts'kyy (2013); Liu et al. (2008)). The reason is thought to be connected to the semi-annual variation of the amplitude of the diurnal tide which has its maxima around equinox.

    The chemical reactions that lead to the emission of electromagnetic radiation by excited sodium atoms at night were first proposed by Sydney Chapman in 1939 (Chapman, 1939) and thus are called the "Chapman mechanism" (See Equations 1 - 3).

$$Na + O_3 \xrightarrow{k_1} NaO + O_2 \tag{1}$$

$$NaO + O \begin{cases} \xrightarrow{f_A \cdot k_2} Na(^2P_J) + O_2 \\ \xrightarrow{(1-f_A) \cdot k_2} Na(^2S_{1/2}) + O_2 \end{cases} \tag{2}$$

$$Na(^2P_J) \xrightarrow{A} Na(^2S_{1/2}) + h\nu \ (589.0/589.6 \text{ nm}) \tag{3}$$

    These reactions yield excited sodium that emits electromagnetic radiation with a wavelength of either 589 nm (D2) or 589.6 nm (D1) (McNutt and Mack, 1963), depending on the exact quantum state of excitation. This state is given by the total angular momentum quantum number $J = 1/2$ or $J = 3/2$. Later, empirical studies found that the D2/D1 line ratio is highly variable

with values between 1.2 and 2.0, which could not be explained by the Chapman mechanism. Laboratory studies showed that this variability could be a result of a dependence on the O$_2$/O ratio. That led Slanger et al. (2005) to the suggestion that the original mechanism needs to be expanded by additional quenching reactions 4 and 5.

$$NaO(A^3\Sigma^+) + O_2 \xrightarrow{k_3} NaO(X^2\Pi) + O_2 \tag{4}$$

$$NaO(X^2\Pi) + O \begin{cases} \xrightarrow{f_X \cdot k_4} Na(^2P_J) + O_2 \\ \xrightarrow{(1-f_X) \cdot k_4} Na(^2S) + O_2 \end{cases} \tag{5}$$

Although, the mechanism of the sodium D-line excitation is now much better understood, there are still uncertainties. These include the values of the branching ratios $f_A$ and $f_X$ that determine the ratio of excited sodium to sodium in the ground state. There has been a lot of research, focusing on ways to derive sodium concentration profiles from sodium D-line nightglow measurements. Xu et al. (2005) showed that using only the reactions of the original Chapman mechanism together with one additional sodium loss reaction and an "effective branching ratio $f$" yields sodium concentrations with uncertainties less than

1 %. Many studies have investigated the "effective branching ratio" and found values between 0.05 and 0.6 (e.g., Hecht et al. (2000); Unterguggenberger et al. (2017); von Savigny et al. (2016); Griffin et al. (2001); Koch et al. (2021)). The sodium





retrieval of this study is based on the results of Koch et al. (2021). By using Lidar measurements to validate the sodium profiles retrieved from Na D-line nightglow measurements with OSIRIS on Odin, they found a branching ratio of $0.064 \pm 0.024$.

A key motivation to find the exact value of the branching ratio is to gain a deeper understanding of the sodium concentrations and their variations in the MLT region. von Savigny et al. (2016) proposed a method to retrieve sodium concentration profiles from night time satellite limb measurements. They show sodium profiles for monthly averaged data for the 30 °S to 30 °N latitude range obtained from the limb measurements with the SCIAMACHY instrument. In the current study, we use the MLT Na profiles retrieved from SCIAMACHY Na nightglow measurements to validate sodium profiles retrieved from OSIRIS night time limb measurements. This paper is structured as follows. In section 2 we give an overview of all the sets of data and the corresponding instruments used in this study. Section 3 describes the Method used to analyze the data. Section 4 shows the results and the main conclusions are presented at the end.

## 2  Data and Instruments

The instruments OSIRIS on Odin and SCIAMACHY on Envisat provide the sodium nightglow measurements, which are the key measurements to obtain sodium profiles. Additionally, we use ozone profiles from the SABER instrument on TIMED and the model NRL-MSIS 00 (See Picone et al. (2002)) provides information on the background atmosphere. Lastly, Na VCDs determined from GOMOS/Envisat stellar occultation measurements were used to validate the results obtained from OSIRIS and SCIAMACHY.

Because SCIAMACHY nightglow measurements only cover the 30 °S to 30 °N latitude range sufficiently and are only available in the years 2003 to 2011, we selected all the other data in a way that they fit in this geographical and temporal range.

### 2.1  OSIRIS on Odin

The first set of sodium nightglow measurements used in this study was provided by the OSIRIS (Optical Spectrograph and Infrared Imager System) (Llewellyn et al., 2004) instrument on the Odin satellite (Murtagh et al., 2002) which was launched on 20 February 2001. The satellite is in a polar, sun-synchronous orbit at an altitude of 600 km with an inclination angle of 97.8°. It completes approximately 15 orbits per day and the satellite has two local equator-crossing times, 6:00 a.m. on the descending and 6:00 p.m. on the ascending leg. Although now, due to orbital drift, they are closer to 6:50 a.m./p.m.. The instrument measures in limb viewing geometry and covers a tangent height range between 5 and 140 km, although a typical measurement in the mesospheric mode only reaches an altitude of 103 to 105 km. The mesospheric measurements have a height sampling of 1.3 - 2 km. The instrument field-of-view is approximately 1 km vertically and 40 km horizontally when mapped onto the atmospheric limb at the tangent point. OSIRIS covers the wavelengths between 280 and 810 nm with a spectral resolution of approximately 1 nm (Llewellyn et al., 2004). The measurements cover the period from 2001 until today, but because we want to compare the results to the results obtained from the SCIAMACHY measurements we only analyze the measurements that fall into the period 2003 to 2011.





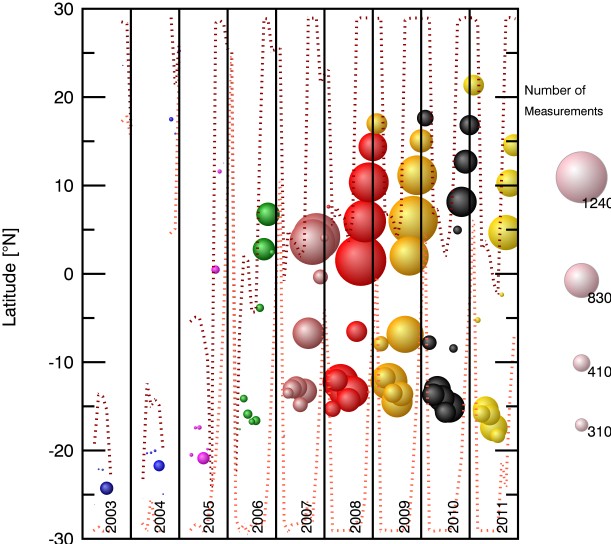

**Figure 1.** Number of OSIRIS mesospheric night mode scans for each month between 2003 and 2011. We show one circle for each month, while the size of the circle represents the number of measurements in this months in the 30°S to 30°N latitude band, the position of the circle corresponds to the average latitude of the measurements in this months. The dotted lines show the minimum and maximum latitude of the available OSIRIS measurements.

Figure 1 shows the number of measurements available in every month during that time period and the mean latitude of the measurements. It is obvious that before 2006 only very few measurements are available and that their mean latitude is shifted

85  to higher latitudes compared to the measurements in later years. To ensure a good comparison we have therefore decided to only include data from the time period 2006 to 2011 into this study. Lastly, to ensure only night time measurements are used for further analysis, we filtered the data for measurements obtained at a solar zenith angle (SZA) higher than $101°$.

### 2.1.1 Pre-processing of OSIRIS data

Several steps have to be taken to make the OSIRIS data usable for the retrieval of sodium profiles. Issues are the variable

90  tangent height grid on which the data is measured and dark currents. With the former we deal by linear interpolation on a standard tangent height grid of 2 km and with the second we deal by offset subtraction. For detailed information see Koch et al. (2021).

Figure 2 shows the resulting spectra from a typical scan after all the necessary steps are taken before the spectra can be used to calculate LER profiles which themselves are the first step to obtain sodium profiles.





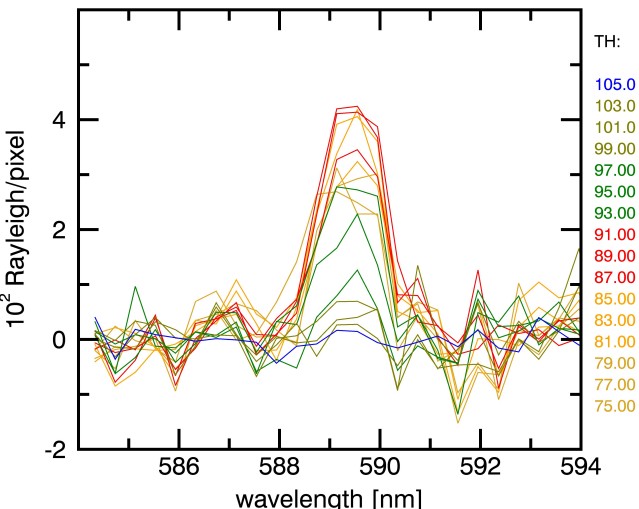

**Figure 2.** Monthly mean OSIRIS Na D-line nightglow spectra for April 2006 after offset and linear fit subtraction. Every colored spectrum refers to one tangent height given on the right. Tangent heights are given in km. For more details about the data pre processing method we refer to Koch et al. (2021).

## 2.2 SCIAMACHY on Envisat

The second data set of sodium nightglow measurements used in this study was measured with the SCIAMACHY (SCanning Imaging Absorption spectrometer for Atmospheric CHartographY) instrument (Burrows et al., 1995; Bovensmann et al., 1999). It was one of ten instruments on the satellite Envisat. Envisat operated from 2002 to 2012 when the European Space Agency (ESA) lost contact to the satellite. So effectively the data covers a period of 9 whole years from 2003 to 2011. Envisat was in a polar, sun-synchronous orbit with a 10 a.m. descending node. The low latitude night time measurements used in this study were carried out at 10 p.m. local solar time. The satellite completed between 14 and 15 orbits per day at an altitude of about 800 km. SCIAMACHY performed measurements in nadir, limb and occultation mode. Its 8 detector channels measured electromagnetic radiation from 214 to 2384 nm, with a wavelength-dependent spectral resolution between 0.2 and 1.5 nm. The 589 nm region, which is important for this study, is covered with a spectral sampling of about 0.2 nm and a sampling ratio of about 2. In this study, only the mesosphere/lower thermosphere limb measurements on the earths' night side (10 p.m. local solar time) were used. In the limb mode the instrument scanned tangentially through the atmosphere in steps of about 3 km from 75 to 150 km tangent height and with a field of view of 110 km horizontally and about 2.6 km vertically. Additionally, we decided to only analyze monthly mean spectra with a mean SNR that is greater than 3. To obtain this value we summed up all the radiances at the sodium D-line wavelengths and divided that signal by the noise. The noise we determined to be the standard deviation of a part of the spectrum where there is no sodium D-line emission and that is equally wide as the part of the spectrum where the signal is located at. We did this for every tangent height and averaged the values.





## 2.3 GOMOS on Envisat

Global ozone monitoring by occultation of stars (GOMOS) on ENVISAT was primarily designed to monitor stratospheric ozone and other chemical species in the earths' atmosphere (Bertaux et al., 2004). While the satellite moves along its orbit, spectral radiances of several stars are recorded with 0.5 s integration time and a vertical resolution better than 1.7 km. GOMOS observed several hundred star occultations every day and it measures in both day and night conditions. The spectral ranges of the GOMOS spectrometers are 250–690 nm, 750–776 nm and 916–956 nm and careful statistical processing of the transmittance data reveals the clear spectral signature of the sodium D2-D1 doublet at 589.16 nm and 589.76 nm respectively (Fussen et al., 2004, 2010). The Na climatology used in this study was obtained by fitting a sodium concentration model to the GOMOS data (Fussen et al., 2010).

## 2.4 SABER on TIMED

To retrieve sodium profiles from measurements of the Na D-line nightglow emission it is required to have knowledge of the corresponding ozone concentration profiles. This study was challenged by the fact that we compare measurements from instruments that measure at different local times (OSIRIS: ca. 6:50 p.m.; SCIAMACHY 10:00 p.m.). To make sure the obtained sodium profiles are comparable, we need ozone profiles from an instrument that provides measurements at different local times. So, we decided to use ozone profiles from the SABER (Sounding of the Atmosphere using Broadband Emission Radiometry) instrument (9.6 $\mu m$ channel, data product: Level 2a, Version: 2.0) (Russell III et al., 2003). The ozone data are, like the OSIRIS and SCIAMACHY data, filtered for the -30° N to 30° N latitude band and for every month of the time span between 2006 and 2011. Additionally, only measurements at a solar zenith angle (SZA) higher than 101° are used for further analysis. In that way, it was ensured that the sun has fully set and only night time measurements are considered. All the remaining measurements are sorted according to local times with a binning of 15 minutes. To select the ozone data that correspond to the OSIRIS and SCIAMACHY measurements, we determine the mean local solar time of all the OSIRIS measurements in one month and then we average all the SABER data in a time interval ranging from that time to 30 minutes after the mean local time.

Using SABER data for the OSIRIS sodium retrieval showed that collocated measurements are not available in every month, between 30° S to 30° N latitude and at around 6:50 p.m. local time. Figure 3 shows all the single measurements of SABER in that latitude band (black) and the mean latitude of the OSIRIS measurements (green). It can be seen that measurements at that local time are only available about every second month. In summary, considering only the months in which OSIRIS measurements in the 30° S to 30° N latitude range with a maximum tangent height of at least 103 km and both SCIAMACHY and SABER measurements are available, leaves 34 months that are suitable for comparison.

## 2.5 Background atmosphere from NRLMSIS-00

Finally, to determine the background atmosphere, temperature, $O_2$ and $N_2$ profiles are taken from the NRLMSIS-00 atmosphere model (Picone et al., 2002). The data was filtered for the latitude band described previously and then we took one profile for every month at 06:00 pm or 10:00 pm for OSIRIS and SCIAMACHY, respectively.





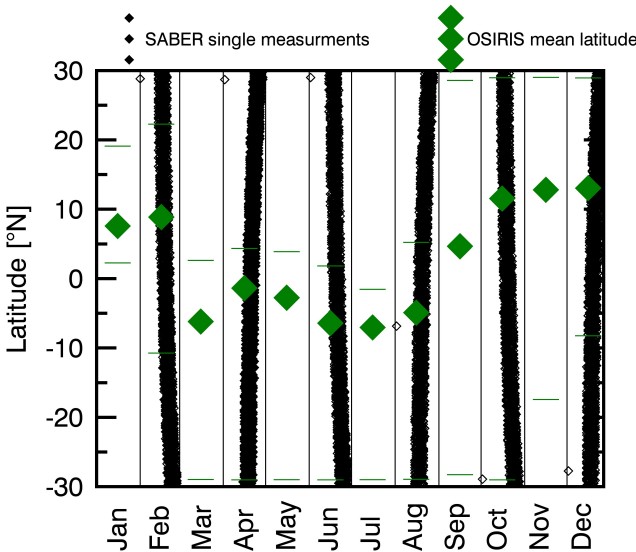

**Figure 3.** Collocation of the mean latitude of the OSIRIS monthly mean spectra (green diamonds) and the SABER measurements (black) that were made between 06:30 and 08:00 p.m. local time in the year 2008. The green horizontal lines indicate the maximum and minimum latitude of the OSIRIS measurements.

## 3 Methodology

To retrieve sodium concentration profiles from VER profiles of the Na D-line nightglow at 589 nm, we used the approach first proposed by Xu et al. (2005). They showed that the Na chemistry is dominated by only 3 reactions. Those are reactions 2 and 3 of the Chapman mechanism and the following additional Na loss reaction:

$$\mathrm{Na + O_2 + M \xrightarrow{k_3} NaO_2 + M} \tag{6}$$

  The steady state assumption leads to the following equation:

$$[\mathrm{Na}]_{\mathrm{ret}} = \frac{\mathrm{VER}/f_A}{k_1[\mathrm{O_3}] + k_3[\mathrm{O_2}][\mathrm{M}]} \tag{7}$$

  Xu et al. (2005) demonstrated that neglecting all other chemical reations leads to Na retrieval errors of less than 1 %. Here, $f_A$ is the effective branching ratio ($f_A = 0.064 \pm 0.024$ , taken from Koch et al. (2021)), and $k_1$ and $k_3$ are reaction rate constants. For $k_1$ we use $1.1 \cdot 10^{-9}\, exp(-116/T)\, cm^3 s^{-1}$ and for $k_3$ $(5.0 \cdot 10^{-30})\, (T/200)^{-1.22}\, cm^6 s^{-1}$. Both values are taken from Plane et al. (2015). For more detailed information see Koch et al. (2021) and von Savigny et al. (2016).

### 3.1 Retrieval Approach and Self-Absorption Correction

In this section we explain how we adjusted the method to use the OSIRIS data instead of the SCIAMACHY data.

To obtain a LER from the spectrum shown in Fig. 2 all the intensities at wavelengths between 588.3 and 591.1 nm are summed


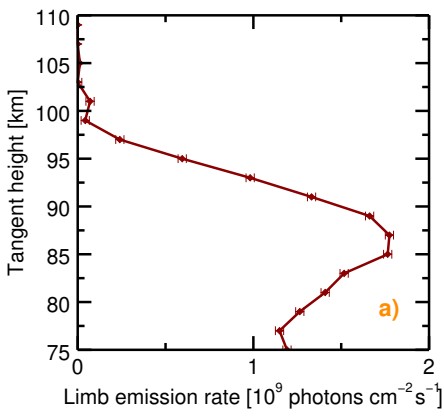
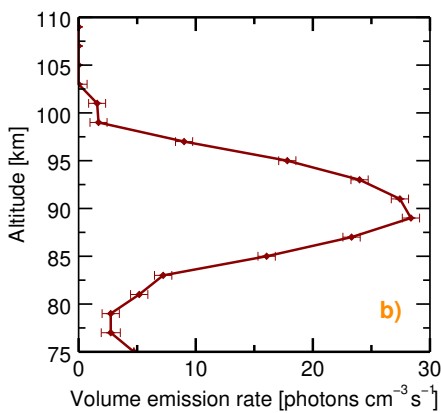

**Figure 4.** a) Sample Na D-line nightglow LER profile for April 2004 determined from OSIRIS measurements. b) sample Na D-line nightglow VER profile that is obtained after inverting the LER profile that was obtained from OSIRIS measurements. It is assumed that there is no emission above 103 km so the VER over 103 km is set to zero.

up. Possible negative non-physical values in the obtained LER profile, resulting from noise, are set to zero because the retrieval does not work with negative LER values. Negative LER values lead to negative VER values and with those the self-absorption

corrections fails to give reasonable Na values. In that way, LER profiles, one for every month, are obtained. Figure 4a shows an OSIRIS LER profile for April 2004. Because the measurements are only available up to 103 km and it is assumed that the LER profiles are zero above that tangent height we add zero values to all tangent heights up to 110 km.

These LERs have to be inverted to vertical VER profiles (Fig. 4b), the first being a function of tangent height and the second being a function of the geometrical altitude. The inversion is explained in von Savigny et al. (2012) and Koch et al. (2021).

The regularization parameter $\gamma$ is chosen to be 1,000. Additionally, the self absorption of the Na D-line emission has to be considered using the same approach as described in Langowski et al. (2016) and Koch et al. (2021).

## 4   Results

This study aims to compare LER, VER and sodium profiles that we have obtained from measurements with two independent (OSIRIS/Odin and SCIAMACHY/Envisat) satellite measurements. Additionally, we want to show the comparison of SCIA-

MACHY and OSIRIS sodium VCDs to those obtained from fitting a sodium density model to GOMOS data (Fussen et al., 2010). The comparison of the LER profiles shown in Fig. 5 of the time interval 2006 to 2011 shows that they are in good overall agreement. Here we show contour plots of SCIAMACHY (upper panel) and OSIRIS (lower panel) monthly LERs for the tangent heights between 80 and 103 km and one profile for each month. The white region in the upper panel is a result of measurements with a SNR below a certain threshold (See Sec. 2.2). The LER peaks lie between 83 and 87 km. von Savigny

et al. (2016) and Unterguggenberger (2017) showed that, at low latitudes, there is a semi-annual cycle in the LERs, with peaks in the spring and fall months. Here, we see the same cycle. Looking at the VER contour plots in Fig. 6 we also see a very


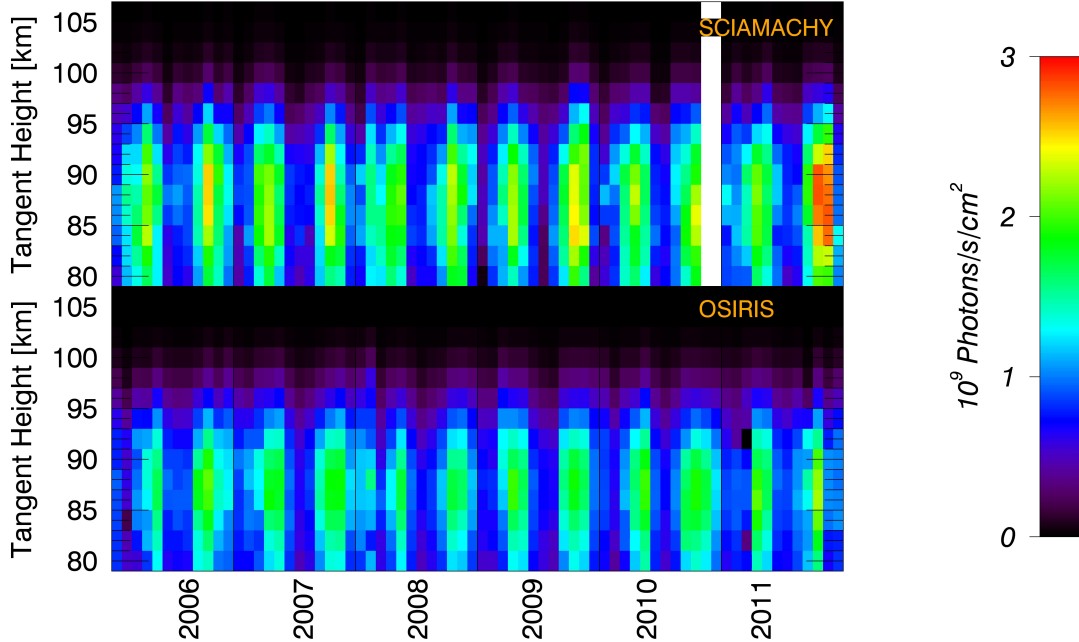

**Figure 5.** Time and altitude dependence of LER calculated from SCIAMACHY and OSIRIS night time limb measurements. Both instruments detect a semi-annual cycle with peaks in the spring and fall months. The white regions are previously selected regions with a low SNR.

good agreement between the SCIAMACHY and OSIRIS VERs. The VERs are shown for altitudes between 80 and 103 km. Although there are more fluctuations than in the LERs, the VER peak is identifiable at around 91 km. Again the semi-annual cycle is clearly visible. To visualize this cycle even better we show the LERs, VERs and also sodium concentration profiles for two tangent heights/altitudes in Fig. 7. This figure shows the LERs (panels a) and b)) and VERs (panels c) and d)) at 83 km and 93 km tangent height and altitude, respectively. For the sodium concentration we chose to show the altitudes of 93 and 95 km. In panel a) and b) the semi-annual cycle of the LERs can be seen very clearly. The panels also show that the SCIAMACHY LER peaks are slightly larger with peak magnitudes of about $2.5 \times 10^9$ photons$/cm^2/s$ than the LER peaks of OSIRIS that are only $2 \times 10^9$ photons$/cm^2/s$, i.e., 20% lower. In Fig. 7 panels c) and d), which show the VER, we find the same semi-annual cycle as in the LERs and again SCIAMACHY shows slightly higher values than OSIRIS. Looking at panels e) and f), which show the sodium concentrations at one altitude, the semi-annual cylcle is not clearly visible but again the sodium concentrations retrieved from OSIRIS measurements are overall lower than those retrieved from SCIAMACHY measurements. Since we can only retrieve OSIRIS sodium concentration profiles for every second month, we interpolated the values in between. Hereby, the green diamonds correspond to the retrieved values.

Figure 8 shows the comparison of the sodium concentration profiles (panel b - SCIAMACHY and c - OSIRIS) as well as the VCDs (panel a). It is obvious that although the shape of the profiles can be slightly different, the values of the sodium concentrations detected by the two instruments agree well with each other. The exact reasons for the high variability, especially





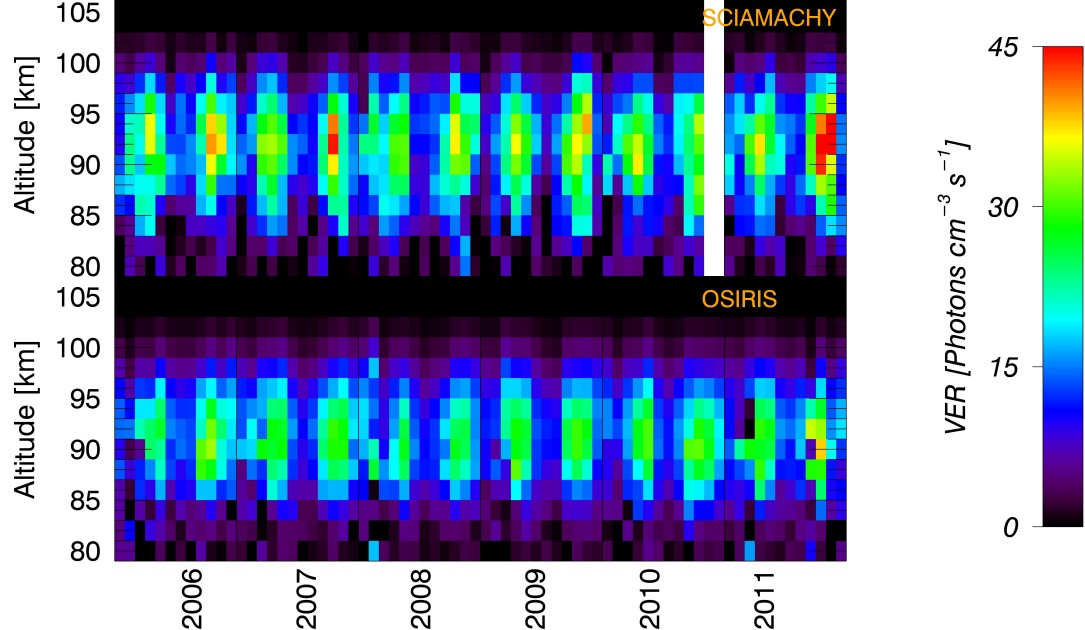

**Figure 6.** VER after inversion of LERs from SCIAMACHY and OSIRIS Na D-line night time limb measurements. Both instruments detect a semi-annual cycle with peaks in the spring and fall months. The white regions are previously filtered regions with a low SNR.

of the SCIAMACHY sodium profiles, are currently not fully understood. In panels c) and b) the sodium concentrations for the altitudes between 80 and 103 km are shown. The sodium peaks are located between 93 and 98 km and varies strongly from

month to month and between the instruments. The white months in panel c) are a result of missing co-located measurements between the SABER (ozone concentration profiles) and OSIRIS measurements. In panel a) we show the VCDs of OSIRIS measurements (green squares), SCIAMCHY measurements (red triangles) and those obtained when fitting a sodium density model to the GOMOS data (pink circles). The latter show a very regular seasonal evolution, which is not as clearly present in the OSIRIS and SCIAMACHY VCDs. Moreover, the SCIAMACHY measurements in most months lead to higher sodium

VCDs than the OSIRIS measurements. That becomes even clearer when looking at the mean values of the sodium VCDs: OSIRIS VCD $= 3.37 \cdot 10^9 \pm 1.07 \cdot 10^9$ atoms/$cm^2$; SCIAMACHY VCD $= 4.25 \cdot 10^9 \pm 1.52 \cdot 10^9$ atoms/$cm^2$; Gomos VCD $= 3.30 \cdot 10^9 \pm 3.54 \cdot 10^8$ atoms/$cm^2$. In Fig. 9 we then show the comparison of sodium profiles retrieved from OSIRIS and SCIAMACHY measurements in four different months of the year 2008 with uncertainties corresponding to each altitude. It can be seen in panel a) and c) that SCIAMACHY yields higher values than OSIRIS and in panel b) the opposite is the case.

Only in panel d) the values of the sodium concentration profiles agree well with each other. We assumed the error of $N_2$ and $O_2$ are 5% of their concentrations, for the mesospheric temperatures we assumed an error of 10 K and for the ozone concentration we used the standard error of the mean. That we determined by dividing the standard deviation of the ozone concentrations at a certain altitude by the square root of the number of measurements which we averaged over to determine the ozone profile.

**Figure 7.** a) LERs of OSIRIS and SCIAMACHY at 83 km and b) 93 km tangent height; c) VERs of OSIRIS and SCIAMACHY at 83 km and d) 93 km altitude and e) sodium concentrations of OSIRIS and SCIAMACHY at 93 km and f) 95 km altitude. Because sodium concentrations from OSIRIS D-line nightglow measurements are only available every second month due two availability of SABER data and in order to help comparability between the two satellite measurements we interpolated linearly between the available sodium concentrations. The actual results are marked with green diamonds, while the interpolated data points have no symbol.

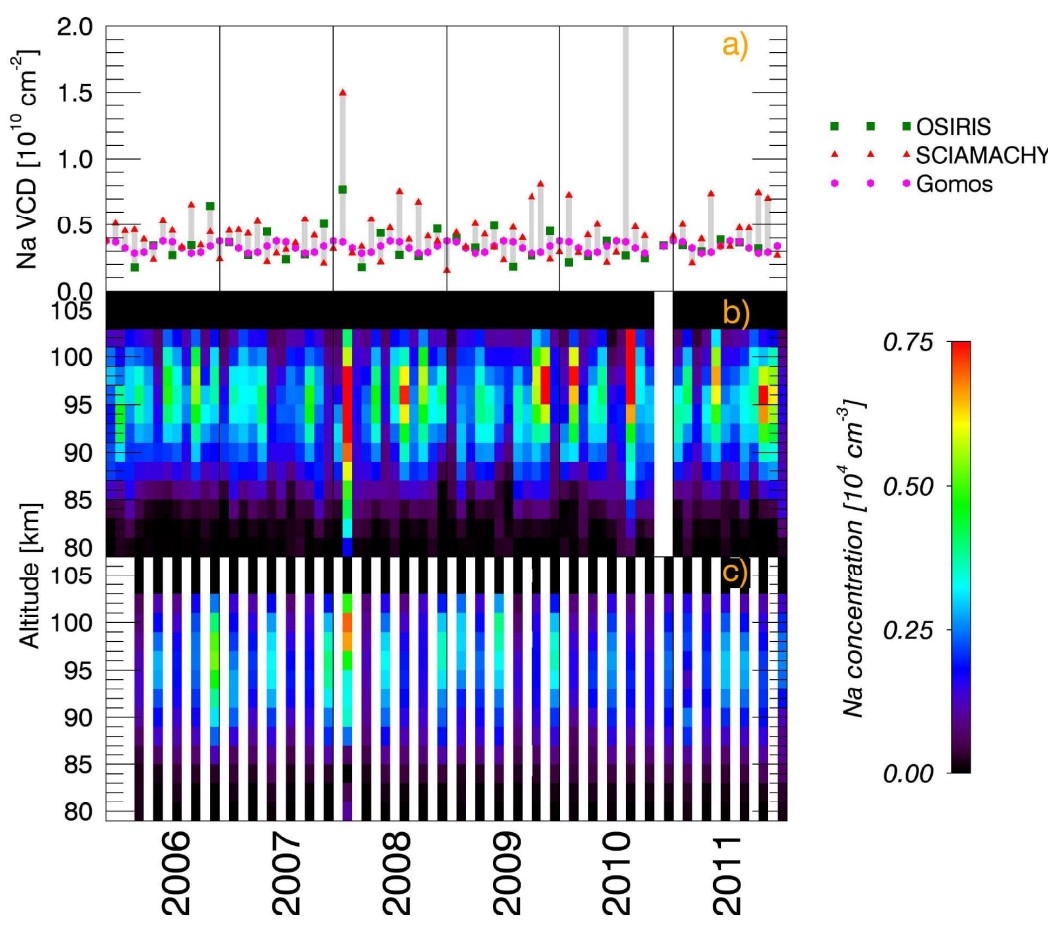

**Figure 8.** a) Sodium vertical column densities obtained from OSIRIS night time limb measurements (green), SCIAMACHY night time limb measurements (red) and a sodium density model (pink) that was fitted to the GOMOS measurements (Fussen et al., 2010). b) SCIAMACHY sodium concentration profiles. c) OSIRIS sodium concentration profiles.

**Figure 9.** Sodium profiles retrieved from OSIRIS (green) and SCIAMACHY (red) mesospheric night time measurements for four month of the year 2008. a) April 2008; b) June 2008; c) October 2008; d) December 2008. The dashed lines show the uncertainties corresponding to each altitude.

We then determined the overall sodium concentration uncertainty by propagation of uncertainty. At an altitude of 95 km it
lies between 2 and 15 %. But in some months the uncertainties can go up to 25 % and above 98 km the sodium uncertainties become very large with values over 200 %.

The differences between SCIAMACHY and OSIRIS become even more obvious when looking at the mean profile of all the available profiles (Fig. 10). Here, we summed up the values of the LERs, VERs and sodium profiles at each tangent





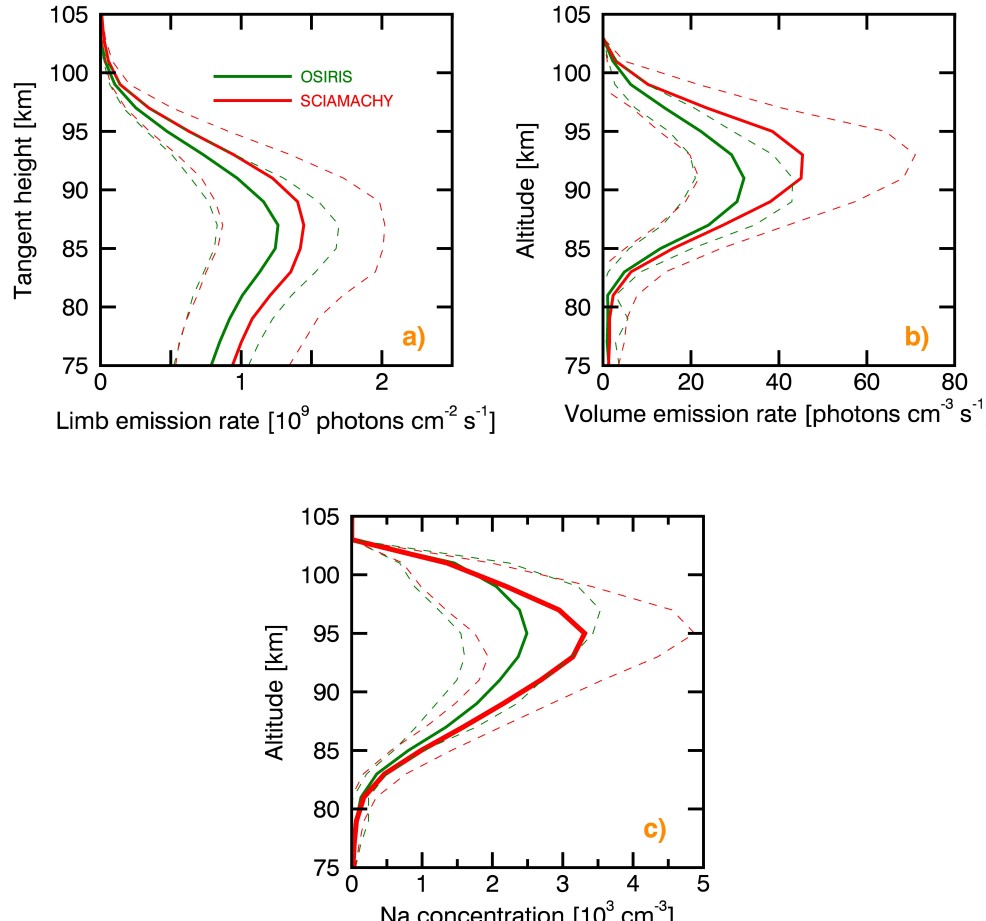

**Figure 10.** Comparison of mean LER profiles a), VER profiles b) and sodium concentration profiles c) of SCIAMACHY (red) and OSIRIS (green) data. To obtain those profiles, we averaged over all available, i.e., 34 monthly averaged Na profiles and 67 LER profiles. The dashed lines show the standard deviation of the LER, VER and sodium concentration profiles of the corresponding altitude.

height/altitude of every available month and then divided the value by the number of months for that we could retrieve

sodium concentration profiles. The mean LER, VER and sodium profiles retrieved from SCIAMACHY (red) are slightly larger than the mean profiles of OSIRIS (green). The dashed lines show the standard deviations for every corresponding tangent height/altitude. For the mean LER profiles the standard deviation is small at high tangent heights and becomes larger between 90 and 95 km. below 90 km it stays large. This is a difference to the standard deviation of the mean VER and sodium profiles. Here the standard deviation is low above and below the peak height and only relatively high in the peak regions at around 90

and 95 km, respectively.





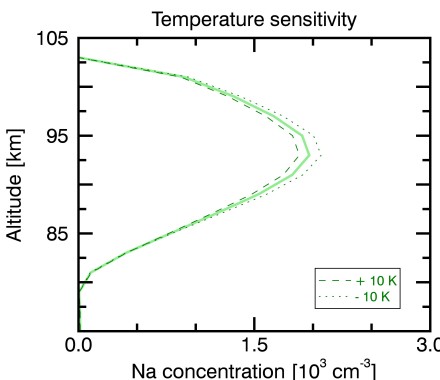
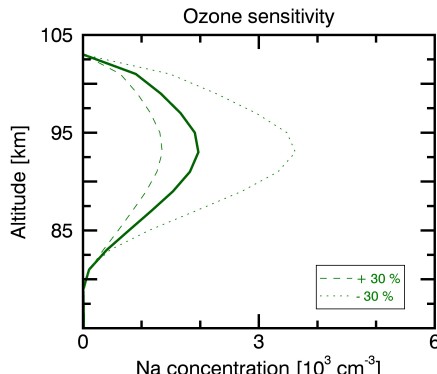

**Figure 11.** Sensitivity of the sodium profile to temperature changes by ± 10 K (left panel) and ozone changes by ± 30 % (right panel).

The reasons for the larger overall values of the SCIAMACHY measurements could be that OSIRIS conducts measurements with a different latitudinal and longitudinal coverage than SCIAMACHY. The first usually only conducts measurements in one hemisphere each month, the latter covers the whole -30° to 30° latitude band. Although we select the profiles of the other parameters accordingly, this difference could still be the reason for the difference in the obtained sodium concentration

profiles. We found out that the parameters with the largest influence are the mesospheric ozone concentration and temperature. In Fig. 11 we show how variations of the ozone concentration and mesospheric temperature influence the corresponding sodium profiles. In the left panel we increased (dashed line) and decreased (dotted line) the temperature and show how the sodium concentration changes in relation to the sodium profile obtained when using the original temperature. Here we see that changing the temperature by 10 K changes the sodium concentration by about 5%. In the right panel we show a similar plot

for the ozone concentration. Here we changed the ozone concentration by 30 %, which corresponds to one standard deviation, and this leads to an increase in the sodium concentration by up to 100 %. Figure 12 shows ozone concentration profiles that are used to retrieve sodium concentration profiles from SCIAMACHY (panel a)) and OSIRIS (panel b)) mesospheric limb night time measurements. The months with no data in panel b) are a result of the differences in latitudinal and longitudinal coverage between OSIRIS and SCIAMACHY. Ozone concentrations range between 1 and $7 \cdot 10^8 / cm^3$ with a peak at 90 km. Also there

is a seasonal variation with peaks in the spring and fall months and low concentrations in the northern hemispheric summer and winter. We found out that low ozone concentrations are related to high retrieved sodium concentrations, especially regarding the SCIAMACHY measurements. All ozone measurements are taken from the SABER/TIMED data base. Ozone is photolysed when the sun is present but recovers very quickly after sun set and is expected to be almost constant during the night (Vaughan, 1982; Schneider et al., 2005; Kreyling et al., 2013). The differences seen in Fig. 12 are probably mainly a result of the different

latitudinal and temporal coverage of OSIRIS and SCIAMACHY. SCIAMACHY covers the whole 30° S to 30° N latitude band, while OSIRIS usually covers only one hemisphere and the SABER measurements are selected accordingly.

Also the local time of the measurements could influence the sodium profiles. As OSIRIS conducts measurements around 6:50 p.m. and SCIAMACHY at 10 p.m.. We select the ozone profiles obtained at those local times to until 30 minutes later and





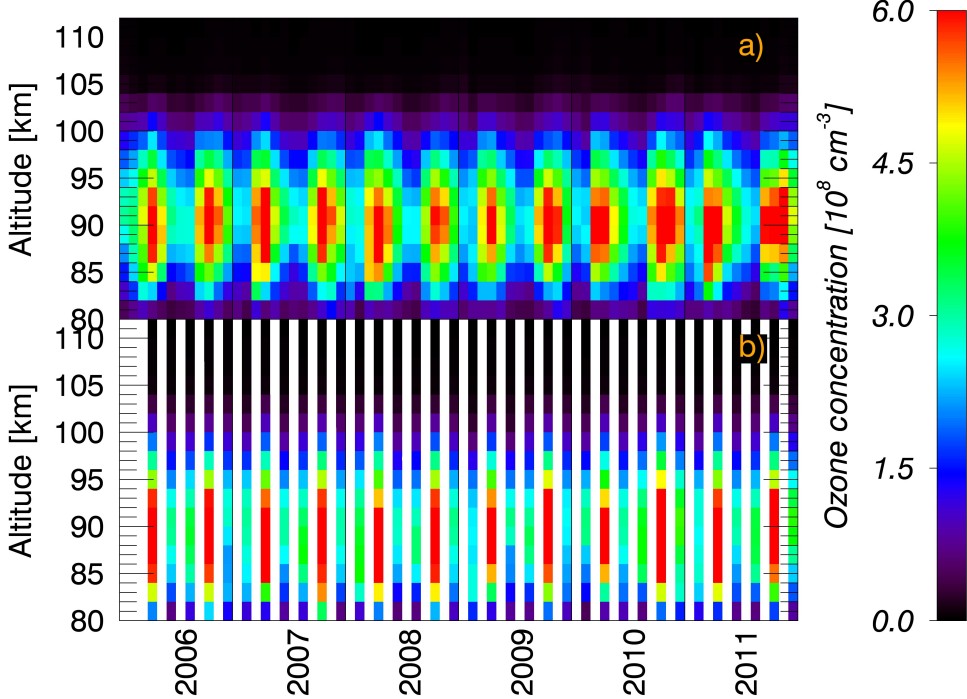

**Figure 12.** Comparison of ozone profiles from SABER measurements used to retrieve sodium concentrations from a) SCIAMACHY and b) OSIRIS mesospheric night time limb measurements.

average over all profiles that fall into this time period. Although we see in Fig. 12 that the variation of ozone between the two

local times is not very big, we have seen in Fig. 11 that small variations of ozone can lead to huge differences of the sodium concentration. In conclusion, the different latitudinal and longitudinal coverage as well as the difference of the local time of the SCIAMACHY and OSIRIS measurements together with the high variability in the mesospheric temperature and ozone profiles can have a big effect on the resulting sodium concentration profiles. And lastly, as already reported in Clemesha et al. (2011), sodium concentrations itself are highly variable during one night. Using lidar measurements to detect sodium layer over Sao

Jose dos Campos (23 °S, 46 °W) they found variations between 2500 and 5000 atoms per $cm^{-3}$.

## 5 Conclusions

In this study, we compared sodium concentration profiles retrieved from Na D-line nightglow measurements from two different instruments, the Optical Spectrograph and Infrared Imager System (OSIRIS) and the Scanning Imaging Absorption Spectrometer for Atmospheric Cartography (SCIAMACHY). We were able to retrieve sodium concentration profiles with uncertainties

between 2 and 15 % and over the period 2006 to 2011 we found 34 months that were suitable for comparison. We found out that, while SCIAMACHY detects slightly larger LERs and VERs, the overall seasonal variation of the three parameters agrees very





well. To retrieve the sodium concentrations we used a branching ratio of $0.064 \pm 0.024$ which was determined by Koch et al. (2021). The same value was calculated by Unterguggenberger (2017) using ozone concentrations profiles from the SABER instrument, and is close to the value found by von Savigny et al. (2016). We were able to show that, as was expected, the
LERs and VERs undergo a semi-annual cycle in the tropics. It was then concluded that the higher concentrations detected by SCIAMACHY could be primarily a consequence of the differences in the spatial and temporal coverage. While SCIAMACHY usually covers both hemispheres in the -30° to 30° latitude band at 10 p.m. local time, the OSIRIS measurements only cover one of the hemispheres each month at around 6:50 p.m. local time. Additionally, the SCIAMACHY mesospheric night time limb measurements, as a result of SCIAMACHYs photodiode array, have a lower SNR than the OSIRIS measurements. We
then analyzed how changing the parameters used to obtain sodium concentration profiles would effect those. We found out that the ozone concentrations have a huge influence on the value of the retrieved sodium. Changing the ozone concentration by 30 %, which corresponds to one standard deviation, changes the sodium concentration by up to 100%. And lastly, we stated that differences in the sodium concentration could be a result of the local time difference between the measurements. While SCIAMACHY measures at a local time of 10 p.m., OSIRIS measurements are carried out at a local time around 6:50 p.m..

*Author contributions.* JK carried out the analysis of OSIRIS data and prepared the manuscript with contributions from all co-authors. CvS carried out the analysis of SCIAMACHY data. AB, NL and CR provided the OSIRIS data and assisted with developing the pre-processing method by providing their knowledge of the OSIRIS data.

*Competing interests.* The authors declare that they have no conflict of interest.

*Acknowledgements.* This project was funded by the *Deutsche Forschungsgemeinschaft* (DPG; DPG grant number: 388042786).



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
