# Peer review of "Comparison of mesospheric sodium profile retrievals from OSIRIS and SCIAMACHY nightglow measurements"

_Atmospheric Chemistry and Physics, 2021_

## Author Comment (AC1)

Replies to Reviewer 2:

We thank the reviewer for his/her constructive comments that helped to improve the manuscript.

1. Reviewer Comment: My primary concern is that the authors should also focus on the science outcome and include one separate section based on the comparison of OSIRIS and SCIAMACHY retrieved sodium density profiles.

**Reply: We are sorry, but we are not sure what the reviewer means by that comment because the results section already provides an in-depth discussion of the comparison between SCIAMACHY and OSIRIS.**

2. Reviewer Comment: The authors have not mentioned about the location of measurements of both SCIAMACHY and OSIRIS (Figures 5). I can understand that these Limb Emission Rates (LERs) are monthly averaged. Are these LERs latitude-longitude averaged? If it is so, what is the scientific basis of doing so? What science outcome can be expected from latitude longitude averaged LERs/VERs?

**Reply: Thanks for pointing out that we haven't mentioned the exact latitude and longitude range of the measurements shown in figure 5. We have added a description to the plot. Those are averaged measurements. They were averaged over the -30 to 30 degree latitude band, all longitudes and over all measurements in each month as already done before with the SCIAMACHY measurements in von Savigny et al. (2016). The averaging is necessary in order to obtain a good enough signal-to-noise. To ensure a good comparability between the two instruments we chose the same latitude region for the OSIRIS measurements.**

3. Reviewer Comment: The authors have discussed on the semi-annual variation in VERs. They should also try to explore other features.

**Reply: We are sorry but we don't really see other obvious features that could be discussed.**

4. Reviewer Comment: The authors have used average $O_3$ density profiles to retrieve sodium density at different altitude. There is a large latitudinal variation in $O_3$ density in the MLT region. Hence from Figure 11 (right panel), it is clear that the retrieved sodium density profile is quite sensitive with $O_3$ variation. How reliable are those sodium density profiles?

**Reply: Thanks for pointing out the sensitivity of sodium profiles to $O_3$ variations. We tried to handle this issue by a proper error analysis. Figure 9 shows sodium profiles with errors that are directly affected by sodium $O_3$ sensitivity.**

5. Reviewer Comment: The authors should discuss how "effective branching ratio" affects the retrieved sodium concentration? They should carry out the sensitivity analysis.

**Reply: We understand that the reviewer is concerned about the effect of the effective branching f ratio on sodium profiles. Equation 7 shows how f affects the retrieved sodium profiles. How a variation of f affects retrieved sodium is already shown in Koch et. al. (2021). They showed sodium concentration profiles, retrieved with branching ratios between 0.05 and 0.21 to compare the resulting profiles to sodium concentrations obtained from ground-based measurements. We have now added a further explanation of this study to the text.**

6. Reviewer Comment: Line 259-260: "…. LERs and VERs undergo a semi-annual cycle in the tropics." The retrieved sodium densities do not show any semi-annual cycle clearly. Why is it so?

**Reply: The reviewer points out that, although the LERs and VERs show a clear semi-annual cycle that is not the case with the sodium concentration. We think that also other studies on that topic came to the conclusion that the semi-annual cycle is not as present in the tropics as in higher latitudes (Langowski et al. (2017), Plane et al. (2015)). And that the prominent semi-annual oscillation in the LERs and VERs has its origin mainly in the seasonal variability of the ozone concentrations. We have added a statement about this to the paper.**

7. Reviewer Comment: Line 262-263: "OSIRIS measurements only cover one of the hemispheres each month at around 6:50 p.m. local time." It appears to me that the LERs from OSIRIS have been measured during twilight time. How do the authors ensure that the contamination from solar background is eliminated?

**Reply: The reviewer wants to know how we made sure that the nighttime limb measurements are not affected by sunlight. To ensure this we used a SZA criterion (SZA must be larger than 101°) as explained in section 2.1**

Minor Comments:

Line 27-28 Is the semi-annual cycle observed in density or emission profiles of sodium? Please mention.

**Reply: It is clearly visible in the sodium emission profiles. This is now mentioned in the paper.**

Line 30: Maintain proper citation style throughout the paper.

**Reply: Citation is corrected**

Line 30-31: "The reason is thought to be connected to the semi-annual variation of the amplitude of the diurnal tide which has its maxima around equinox". The authors should provide references. Does semi-diurnal tide have any role?

**Reply: We have now provided reference. We have to admit that we are not sure about semi-diurnal tides in the sodium layer.**

Line 40: "Laboratory studies showed that this variability could be a result of a dependence on the $O_2$/O ratio" Please provide references.

**Reply: Reference is now provided**

Line 90: Why is the linear interpolation chosen to deal with variable tangent height?

**Reply: Because it is the easiest way of interpolating and we think it leads to sufficient results.**

References:

Langowski, M. P., Savigny, C. V., Burrows, J. P., Fussen, D., Dawkins, E. C., Feng, W., Plane, J. M., and Marsh, D. R.: Comparison of global datasets of sodium densities in the mesosphere and lower thermosphere from GOMOS, SCIAMACHY and OSIRIS measurements and WACCM model simulations from 2008 to 2012, Atmos. Meas. Tech., 10, 2989–3006, https://doi.org/10.5194/amt-10-2989-2017, 2017.

Plane, J. M., Feng, W., and Dawkins, E. C.: The Mesosphere and Metals: Chemistry and Changes, Chem. Rev., 115, 4497–4541, https://doi.org/10.1021/cr500501m, 2015.

von Savigny, C., Langowski, M. P., Zilker, B., Burrows, J. P., Fussen, D., and Sofieva, V. F.: First mesopause Na retrievals from satellite Na D-line nightglow observations, Geophys. Res. Lett., 43, 12,651–12,658, https://doi.org/10.1002/2016GL071313, 2016.

---

## Author Comment (AC2)

Replies to Comments of Reviewer 1:

We thank the reviewer for his/her constructive comments that helped to improve the manuscript.

1.  Reviewer Comment: Some clarifications are needed when it comes to reactions and reaction coefficients:
You first introduce the basic Chapman mechanism (your equations 1-3). You then introduce the extension by Slanger (your equations 4-5) in order to explain the variable ratio of D1/D2 observed in the nightglow. In order to make this scheme consistent you should make clear that equation 1 describes specifically the production of NaO(A), not NaO in general.

   **Reply: We clarified that reaction 1 produces NaO(A) and not NaO in general**

Reviewer Comment: You should be consistent when referring to the effective branching ratio. In section 1, $f_A$ is the branching ratio in the original Chapman mechanism and you refer to the effective branching ratio introduced by Xu et al. (2005) as f. In the section 3, however, you refer to the effective branching ratio as $f_A$ (line 152).

   **Reply: We now refer to the effective branching ratio as "f" and to the two branching ratios in the original and the extended mechanism as fA and fX, respectively.**

Reviewer Comment: In line 146-147, you state "These are reactions 2 and 3 of the Chapman mechanism…". I suppose that this should be reactions 1 and 2 instead.

   **Reply: Thanks for pointing out that we referred to the wrong equations. This is now corrected.**

Reviewer Comment: In equation 6 you introduce the reaction coefficient $k_3$. However, a coefficient $k_3$ is already used in equation 4. So one of these coefficients needs to be renamed.

   **Reply: We renamed the reaction rate coefficients of equations 4 and 5 to make sure that there is no ambiguity**

2.  Reviewer Comment: It would be good to add some clarification about the OSIRIS database used for this study. While it can be discussed whether Figure 1 is the most instructive way of providing an overview of this database, I certainly appreciate the creativity that went into developing this figure. Still, one clarification should be added: In line 87 you state that only solar zenith angles (SZA) larger than 101 degrees are used in the analysis. However,as I understand it, no restriction in SZA is applied in Figure 1. What fraction of the data in Figure 1 remains once the SZA limit of 101 degrees is applied?

   **Reply: We updated figure 1 so that it now shows only measurements with SZA > 101° and changed the caption accordingly. We also made clear that it only shows measurements that were carried  out on the ascending leg of the satellites flight path.**

3.  Reviewer Comment: A discussion is needed about possible biases introduced by the data analysis: You set negative radiance values to zero. Please discuss how much this can affect the mean values used in the analysis.

   **Reply: The reviewer points out that setting negative values to zero could lead to a positive bias  in the retrieved sodium concentrations. We admit that this is true but we are not sure**

how to quantify this effect because the retrieval method only allows for values greater than or equal to zero. This is the case, because the self-absorption correction requires an iterative retrieval of the Na concentrations. If negative LERs would be allowed, the Na concentrations will or may become negative and the retrieval stops. For this reason, we have to reject the negative LERs. We think we can accept the bias because negative values never occur in the peak regions but only at high altitudes.

Reviewer Comment: Variables like ozone, temperature etc. do not enter the retrieval relationships linearly  Still, your retrieval is based on applying the retrieval  relationships to monthly averages of the individual variables. Please discuss how much the nonlinearity may affect your monthly mean sodium results.

Reply: Thanks for this comment. We tested how the non-linearity of the sodium sensitivity to ozone affects the overall sodium profiles by retrieving sodium profiles with all the individual ozone profiles before averaging those to obtain the monthly ozone profile. We then took the average over all the resulting sodium profiles and compared this to the sodium profiles that was obtained with the monthly ozone profile. We found out that the effect in most months is +- 5% of the sodium peak concentration. And only a few months in which the effect is larger than 20%. We added a discussion on this to the paper.

Reviewer Comment: In lines 205-211 you discuss the absolute error of the retrieval. You list uncertainties of $N_2$, $O_2$, $O_3$ and temperature as contributing factors. However, you do not mention uncertainties in the absolute calibration of OSIRIS an SCIAMACHY, which I assume can be critical. Please discuss this.

Reply: The VER error also affects the total sodium density error. This is now mentioned in the text.
The absolute calibration error of OSIRIS is estimated to be between 5 and 10 percent and for SCIAMACHY between 2 and 4 percent (For more information see the SCIAMACHY read me file for the Level 1b version 8.0X dataset). To estimate how  this affects the sodium retrieval we changed the LERs by +- 10 percent. This leads to a change in the sodium concentration of about 15 %. This is now mentioned in the text.

4.  Reviewer Comment: I am confused about the units of the limb emission rate. e.g in Figure 4, Figure 5, Figure 7 and in the text. I suppose that the correct unit should be that of a radiance (photons cm$_{-2}$ s$_{-1}$ sr$_{-1}$) rather than photons cm$_{-2}$ s$_{-1}$.

Reply:  The units are correct, as we carried out a conversion from limb radiance to limb emission rate.

5.  Reviewer Comment: The very high monthly averages ("outliers") in the SCIAMACHY retrievals in early 2008 and mid 2010 are astonishing (Figure 8). You state that the exact reasons are currently not fully understood. I understand that this may be beyond the scope of this paper. Still, it should be possible to provide some basic analysis. Are sodium concentrations during these entire months systematically high? Or is there a limited set of sodium profiles with very high (erroneous?) values that strongly affect the monthly average?

Reply: As stated in the results section we believe that the very high values of the sodium concentrations are a result of very low ozone concentrations because those outliers only

**occur in months were ozone concentrations are lowest. As we showed small variations of the ozone concentrations lead to large variations in the sodium concentrations.**

6.   Reviewer Comment: As a possible reason for the deviation between SCIAMACHY and OSIRIS you list different latitudinal sampling (lines 221-223). I suppose that this could easily be checked by selecting of a subset of the datasets and making sure that averaging is done over consistent latitudes for both instruments.

**Reply: We are very thankful for the idea to just use a subset of the SCIAMACHY data for the analysis. We did the analysis only with SCIAMACHY measurements that fall in the same latitude range as the OSIRIS measurements in the corresponding month. Unfortunately this leads to more outliers in the SCIAMACHY sodium concentrations. This is attributed to SCIAMACHY'S low signal-to-noise ratio. So, a large amount of SCIAMACHY measurements needs to be averaged to obtain spectra that are suitable for sodium retrieval. See von Savigny et al. (2016)**

Minor comments:

**Thanks for all the minor comments. The errors have been corrected and what needed clarification was clarified**

References

von Savigny, C., Langowski, M. P., Zilker, B., Burrows, J. P., Fussen, D., and Sofieva, V. F.: First mesopause Na retrievals from satellite Na D-line nightglow observations, Geophys. Res. Lett., 43, 12,651–12,658, https://doi.org/10.1002/2016GL071313, 2016.